# Willingness to Use Male Contraceptive Pill: Spain-Mozambique Comparison

**DOI:** 10.3390/ijerph20043404

**Published:** 2023-02-15

**Authors:** Piedad Gómez-Torres, Ana C. Lucha-López, Amber Mallery, Guillermo Z. Martínez-Pérez, Germano Vera Cruz

**Affiliations:** 1Department of Physiatrics and Nursing, Faculty of Health Sciences, University of Zaragoza, 50009 Zaragoza, Spain; 2Research Group Sector III Heathcare (GIIS081), Institute of Research of Aragón, 50009 Zaragoza, Spain; 3Unidad de Investigación en Fisioterapia (UIF), University of Zaragoza, 50009 Zaragoza, Spain; 4African Women’s Research Observatory, 08035 Barcelona, Spain; 5Department of Psychology, University of Picardie Jules Verne, 80000 Amiens, France; 6Centre de Recherche en Psychologie: Cognition, Psychisme et Organisations UR 7273 CRP-CPO, University of Picardie Jules Verne, 80025 Amiens, France

**Keywords:** willingness to use male contraceptive pill, Spain, Mozambique

## Abstract

Previous studies have suggested that social and cultural factors significantly influence people’s willingness to use the male contraceptive pill, which is in relatively advanced development. The present study aims at comparing Spanish and Mozambican participants level of willingness to take a male contraceptive pill. Factorial designed scenarios were used to collect data on the two population samples (Spain = 402 participants; Mozambique = 412 participants). One-way analysis of variance (ANOVAs) were performed comparing the average scores of Mozambique and Spain at the levels of each modelled factor: The cost of the pills (30 €/USD 20 for 3 months vs. free); Efficacy (99% vs. 95%); Side effects (none, mild and severe); Context (disease, condom abandonment and diversification of contraceptive methods). The two groups found significant differences in the scores for each of the four factors, in light of the socio-cultural differences between the two countries. In the Spanish sample, the main factor affected the willingness to use male contraceptive pill (MCP) were the side effects, while for Mozambican men it was the context. Along with technological change, an ideological-social change in gender roles is required to ensure equity in contraceptive responsibilities and the participation of men at all socio-demographic levels in reproductive health.

## 1. Introduction

Gender is a determining factor of health inequities [1]. The pharmaceutical industry has been focused on developing birth control methods for women since the 1960s [2]. Men do not have the same reproductive health opportunities as women with regard to contraception, implying gender inequity in reproductive rights and access. With the objective of providing men with a new method of contraception and expanding the range of contraceptives available for men, the male hormonal contraceptive pill is being developed. Recent clinical trials of a male contraceptive pill have reported very promising results [3,4] indicating its potential availability in the relatively near future. The availability of a reversible and efficient male contraceptive pill has the possibility to generate more equitable decision making when it comes to family planning and reproductive health. As well as address the gender burden of pharmaceutic contraceptives on women. However, to have a positive impact both in ushering a “new revolution” in the contraception field and in the reducing the gender-based responsibility of contraception, the use of the male contraceptive pill must be widely accepted.

In the Global North, the contraceptive prevalence rate for men and women has been higher over the years than in the countries of the Global South [5]. In recent years, it should be noted that although contraceptive use is low in sub-Saharan Africa compared to other regions, Mozambique is the country with the highest increase in modern contraceptive use for men and women [6]. In Mozambique, the contraception prevalence among women of reproductive age (15–49 years) in recent years was estimated to be less than 30% [5]. On the other hand, government spending on family planning is one of the lowest among low- and lower-middle-income countries at only 7% of spending, thus relying heavily on donors, leaving an unmet need for family planning [7].

Meanwhile, the only fully reversible reliable male contraceptive available is the condom. In general, the use of condoms in these two countries is subject to sexual encounters in non-stable relationships, since men in stable relationships are more reluctant to continue using them in the advancement of relationships [8]. Many men are reluctant to use condoms because they feel that they reduce their sexual pleasure [9]. Condom use in Spain has fallen in the last decades, going from 84% use in 2002 to only 75% use in 2018 [10,11]. This decrease in condom use has led to a 26% increase in sexually transmitted infection in Spain, leaving a similarly high HIV rate between Spain and Mozambique [12,13]. In Mozambique, the prevalence of the use of condom as contraceptive mean was less than 8% [6]. In 2010, the Mozambican government launched a national program to make contraceptive methods for women including “the pill” available [14]. While contraceptive use in Spain has decreased, in Mozambique it has increased significantly [5,14,15].

As for the consequences of an unwanted pregnancy, not only must it be considered that it is women who become pregnant, but often, the gender expectations in many countries and cultures place women in the role of the main caregivers of children. Generally, in the context of high-income countries, men’s contraceptive responsibility is often related to women’s contraceptive decisions [16]. In Mozambique, recently, a minority of men indicated that contraceptive responsibility fell on both members of the sexual partner [14]. Research suggests that Mozambican men determine the contraceptive decisions of the partnership in regard to their perceived masculinity [17], which is why there is an unmet need for family planning. On the other hand, in Spain, in 2019, 91% of men indicated that contraceptive responsibility should be shared by both partners of the couple [18].

In terms of fertility, in Mozambique about 50% of Mozambican women between the ages of 15 and 19 are pregnant or already mothers [19]. In addition, half of the premature deaths among women between the ages of 15 and 24 are related to pregnancy, childbirth or abortion [14]. In Spain, only 3% of women between the ages of 15 and 19 are pregnant or already mothers, and 79.2% of women between the ages of 25 and 29 have not yet had children [20].

Spain and Mozambique are two countries with many differences in their sociodemographic characteristics (See Table 1) [21]. Mozambique is a sub-Saharan country on the south-eastern coast of Africa with a population of 31 million, low income, a high risk of poverty, a high rate of unwanted pregnancies, and an average literacy rate of the population of 72.6% [21]. Spain, on the other hand, is a country in southern Europe with a population of 47 million inhabitants, of medium-high level of economic income, a low rate of unwanted pregnancies and a high literacy rate [21]. Examining the social context, in Spain at 15 years of age a girl is viewed as a child, while in Mozambique girls of the same age are viewed as women [17]. Spanish women have access to and social acceptance of contraception and their in decisions regarding contraception [11]. Contrarily, in Mozambique, family size is linked to masculinity and contraceptives to promiscuity among women [17]. Furthermore, in Mozambique a woman’s future is largely dependent on having children after she leaves school [22], while in Spain young women attend school/higher education to prolong the social expectations of adulthood [23]. 

Traditional values/roles regarding male-female relationships are much more prevalent in Mozambique than in the Global North [24,25]. It is believed that this may affect the willingness to use a male contraceptive pill (MCP). Often what is expected of men in countries in sub-Saharan Africa, such as Mozambique, are playing into the social perception of appearing virile and showing sexual prowess. In Mozambique, manhood itself is socio-culturally grounded in a person’s ability to procreate [17]. In addition, the notion of being a father and continuing the family lineage carries power and status within the community and cultural setting [17], which may have some impact on a man’s desire to accept MCP in Mozambique. Conversely, women carry the responsibility to “conceive” their husbands’ children, a social mission highly associated with a women’s worth from a sociocultural context [17]. Mozambican women often rely heavily on marriage to ensure economic security and safety [17]. With this in mind, MCP has the potential to shift the perception of conception responsibility in Mozambique. 

While in Spain the manifestations of the traditional model of male socio-cultural dominance over women still exist, women are still accepting care roles that should not be the exclusively responsibility of women [26]. As stated above, many male individuals in high-income countries justify their reluctance to use condom for contraception with the fact that the use of a condom is, according to them, associated with diminished pleasure sensation [9]. Research has shown acceptability of condom usage is higher within Spain as compared to Mozambique [13]. The cultural implications of condoms in environments where they are associated with promiscuity, and where religious aspects interfere with the ethics of sexual practices, are associated with this lower acceptance in Mozambique [17].

The unmet need for comprehensive family planning for both sexes indicates that a new and efficient method of contraception for men is necessary to give them an opportunity to contribute more than they currently do on this matter. More studies must be carried out to try to understand in different global contexts, to what extent male individuals will be willing to use such a new method of contraception and what are the health, social, cultural, and economic factors likely to influence that willingness.

The present study aims at assessing the existing differences between Spanish and Mozambican participants regarding the willingness to use a male contraceptive pill. Particularly, the current study aims at examining the two groups willingness to use male contraceptive pill under different conditions (factors): the pill’s cost, the pill’s efficacy, the pill’s side-effects, and the pill’s use context.

In the current study, willingness (to what extent an individual wishes, desires, or wants something) is processed in light of the Information Integration Theory and Functional Measurement which is a psycho-cognitive approach that tries to account, in human beings, for the processes of evaluating information (perceived stimuli) and integration of valued information to form a judgment [27].

Research on willingness to use male hormonal contraceptives prior to the data presented in this analysis included data from Spain only in a multicenter study published in 2005 [28], without previous studies in Mozambique. To our knowledge, no study has been conducted comparing a Southern African country and a Global Northern country in this matter, using the method applied in the current study. Thus, this study may contribute to a better understanding of the factors that may influence men’s attitudes towards the use of male contraceptives pill, both in Global North such as Spain and Global South such as Mozambique.

## 2. Methods

Comparison of data collected in two population groups of a descriptive cross-sectional survey. Data from the first population group was collected in 2018 in Mozambique, and data from the second population group was collected in 2019 in Spain. Both studies used the same data collection methodologies and similar recruitment and sampling procedures.

### 2.1. Participants

Participants in the Mozambique study were men who were living in three provinces of Mozambique: Maputo, Sofala and Nampula. Participants were interviewed by six research assistants between April and October 2018. The participants of the study carried out in Spain were men who were living in the community of Aragon (Spain). Participants were recruited by a doctoral student in Health Sciences. In both studies, participants were recruited, unpaid volunteers. The non-response rate in Mozambique was 40% while in Spain it was 23.7%. The socio-demographic data analyzed in Spain do not show any differences between men who responded to the survey and those who did not.

### 2.2. Inclusion and Exclusion Criteria

Inclusion criteria: men aged between 18 and 49 years in Mozambique and men aged between 15 and 49 years in Spain, regardless of their country of birth, ethnic affiliation, educational level, sexual orientation, gender identity and socioeconomic status, and who gave their informed consent to participate in the survey. All individuals who did not meet the inclusion criteria were excluded from the study; and those individuals who did not meet the inclusion criteria and who did not agree to participate in this study.

### 2.3. Sampling and Recruitment

In Mozambique: Research assistants positioned themselves on the main streets in towns, cities and villages in the three provinces (Maputo, Sofala and Nampula) and approached one in three men who passed by. In Spain: participants were randomly sampled in three phases. First, a randomization of the districts of the city of Zaragoza where the recruitment would take place. Second, stratified random sampling of recruitment points where the recruitment would take place was made, according to the size of the population of men 15–49 years-old living in each municipal district of the city. Then, a sample size proportional to each municipal district was calculated. Finally, an on-site randomization was carried out at each recruitment point. The third out of every three men who passed through the area both in the morning and in the afternoon on alternate days was invited to participate.

### 2.4. Measures

In order to make the data comparable, the same data collection and measurement tool was used in both countries, adjusting the cost of the method according to the country. The instrument consists of 36 different scenarios in which at the end of each one the question is asked on a Likert scale of 0–10 points (from certainly not–0 to certainly yes–10) if the participant was the male partner, to what extent would they agree to use these pills (describing the effectiveness of the pill, side effects, cost and context in each of the scenarios/vignettes) to prevent your female partner from becoming pregnant. The 36 scenarios were created by orthogonally crossing the factors: Cost (free vs. 30 € for 3 months); Efficacy (95% vs. 99%); Side effects produced (none, mild or severe); and Context of use where three reasons were considered: medical, equality and method diversity. In the medical condition, the couple could no use oral contraceptives for health reasons. In the equality reasons, the female partner felt that she was the one who had been taking the pill, it was time for her partner to share the burden of contraception. In the condition of diversity methods, the couple had been using condoms but was now considering switching to the MCP. Note that the cost of the pill for Mozambique was converted to local currency and to the corresponding cost of living.

For the design of the data collection instrument, pilot research was conducted in Mozambique and in Spain.

Sociodemographic variables (age, socioeconomic level, educational level, religion) were recorded after the scenarios/vignettes in both countries. The degree of religiosity was also measured (with the Hoge intrinsic religious motivation scale [29]).

### 2.5. Procedures

In both studies, prior to each data collection session, participants signed informed consent and received a synopsis of MCP research (the chemical content of the “future” pill, its functioning mechanisms, possible side effects, etc.) from the principal investigator. The researchers answered questions and clarifications requested by the participants during data collection. Participants took between 45 and 60 min to respond to all scenarios in both countries. After answering the scenarios, they answered additional questions about their age, sex, level of education, level of religiosity, etc. In Mozambique, all instructions and materials were submitted in Portuguese and in Spain they were submitted in Spanish.

### 2.6. Ethics

For the study conducted in Mozambique, ethics approval was obtained from the Ethics Committee of the Eduardo Mondlane University in Maputo. For the study carried out in Spain, which includes this analysis, ethics approval was obtained from the Research Ethics Committee of the Autonomous Community of Aragon (Spain) (C. P.–C. I. PI19/192).

### 2.7. Statistical Analysis

The scores of willingness to take a male contraceptive pill were converted to numerical values from 1 to 11 and all subsequent analyses were based on these values. In order to compare the groups on the factors that may determine the use of the male contraceptive pill in each of the populations, we first measured the 36 responses for each vignette (with the 4 factors that make up each of them). One-way ANOVAs were performed comparing the average scores of Mozambique and the average scores of Spain at the levels of each factor: level 30 € for 3 months and free level for the Cost Factor; level 99% and level 95% for the Effectiveness Factor; levels none, mild and severe for the Side Effects Factor; and levels medical condition, female partner and diversification of methods for the Context Factor. The significant threshold was set at 0.05.

## 3. Results

Participants in the Mozambique study were 412 men aged 18 to 47 years (M = 27, SD = 4.3 years) and the participants of the study carried out in Spain were 402 men aged 15 to 49 years (M = 30.53, SD = 7.89 years). The sociodemographic characteristics are shown in Table 2.

The Spanish and the Mozambican mean scores of willingness to use male contraceptive pill at each level of each of the four factors are shown in Table 3. The first four columns of the table show the four factors included in the study and their respective levels. In the following columns, the mean scores for each country are presented and show the differences by comparing the means of the two countries for each factor and level. The highest the mean, the highest the willingness. The average of the Spanish men’s responses to the 36 vignettes were found to be higher than the averages of the Mozambican men’s responses in all the factors analyzed that may influence the MCP up taking.

Table 4 presents the two countries comparisons (ANOVA summary) at each level of each factor. 

Regarding the cost factor: Country has a significant effect on the participants’ willingness score associated with the 30 € cost for the male pill [*F*(1) = 42.6, *p* < 0.001, η^2^_p_ = 0.56]; Spanish men have higher score associated with the 30 € pill cost (M = 6.73, SD = 1.53) than Mozambican (M = 3.35, SD = 1.57). Country as a significant effect on the participants’ willingness score associated with the free-cost for the male pill [*F*(1) = 22.72, *p* < 0.001, η^2^_p_ = 0.40]; Spanish men have lower score associated with the free-cost of the male pill (M = 6.9, SD = 1.65) than Mozambican man (M = 4.13, SD = 1.83).Regarding the effectiveness factor: Country has a significant effect on the participants’ willingness score associated with 99% effectiveness of the male pill [*F*(1) = 33.0, *p* < 0.001, η^2^_p_ = 0.49]; Spanish men have higher score associated with the 99% effectiveness (M = 7.42, SD = 1.57) than Mozambican man (M = 4.07, SD = 1.91). Country has a significant effect on the participants’ willingness score associated with the 95% effectiveness of the male pill [*F*(1) = 34.6, *p* < 0.001, η^2^_p_ = 0.50]; Spanish have lower score associated with the 95% effectiveness of the male pill (M = 6.20, SD = 1.34) than Mozambican man (M = 3.41, SD = 1.50).Regarding the context factor: Country has a significant effect on the participants’ willingness score associated with the medical context to take the pill [*F*(1) = 12.97, *p* = 0.0016, η^2^_p_ = 0.37]; Spanish men have lower score associated with medical context (M = 7.46, SD = 1.45) than Mozambican (M = 5.03, SD = 1.84). Country has a significant effect on the participants’ willingness score associated with the equity in the couple to take the pill [*F*(1) = 45.17, *p* < 0.001, η^2^_p_ = 0.67]; Spanish have higher score associated with equity context (M = 6.78, SD = 1.57) than Mozambican man (M = 2.90, SD = 1.23). Country has a significant effect on the participants’ willingness score associated with the context to stop using condom to take the male pill [*F*(1) = 24.17, *p* < 0.001, η^2^_p_ = 0.52]; Spanish men have higher score associated with the context to switching from condom to taking the male pill (M = 6.19, SD = 1.54) than Mozambican man (M = 3.29, SD = 1.34).Regarding the side-effects factor: Country has a significant effect on the participants’ willingness score associated with none side-effects of the male pill [*F*(1) = 32.24, *p* < 0.001, η^2^_p_ = 0.59]; Spanish men have lower score associated with none side-effect of the male pill (M = 8.36, SD = 0.94) than Mozambican men (M = 5.49, SD = 1.48). Country has a significant effect on the participants’ willingness score associated with light side-effects of the male pill [*F*(1) = 81.1, *p* < 0.001, η^2^_p_ = 0.79]; Spanish men have higher score associated with light side-effect of the male pill (M = 6.91, SD = 0.83) than Mozambican (M = 3.36, SD = 1.08). Country has a significant effect on the participants’ willingness score associated with severe side-effects of the male pill [*F*(1) = 73.47, *p* < 0.001, η^2^_p_ = 0.77]; Spanish men have higher score associated with severe side-effect (M = 5.16, SD = 0.81) than Mozambican man (M = 2.38, SD = 0.79).

The two groups have significant differences in scores with respect to each of the four factors presented in the scenarios (Table 3).

Figure 1 displays the main patterns of data that correspond to the comparison Spain-Mozambique. The three curves correspond to the three levels of the context factor. 

As Figure 1 suggests, in the Spanish sample, the main factor that impacted the willingness to use MCP were the side effects, while for Mozambican men it was the context, specifically when the female partner could not take a hormonal contraceptive due to a medical problem.

## 4. Discussion

The present study found significant differences in the scores for each of the four factors of willingness to use MCP between Spanish and Mozambican men, in light of the sociocultural differences between the two countries. Results suggest that the willingness to take a MCP is higher among Spanish men compared to Mozambican men. This finding may be associated with differences in sexual health education, gender roles, and sociocultural influence such as stigma, religious beliefs and/or social norms [24,30,31]. Mozambican men reported relatively higher willingness to use MCP when the female partner was unable to take a hormonal contraceptive because of a medical problem. This particular finding suggest that the majority of Mozambican participants are willing to grant an exception to their traditional view of gender roles associate with contraception responsibilities [25]. In addition to this reason, further research would be needed to know if there is other reasons likely to drive Mozambican men into using MCP, given that the willingness to use the pill seems to have a high social component [17]. It is possible that with comprehensive sex education and a social activism [31,32], they would be more willing to use this kind of contraceptive. Likewise, the difference in willingness to take a MCP between the two countries could be related to the greater acceptance of the use of the method by men’s referents and their social environment, as is the case with the condom [13].

Given that in Spain the level of religiosity is lower among the men surveyed, this could be one of the factors most conducive to making the pill attractive on the market in this country. Research in Mozambique has found a significant association with religions influence on perceived gender roles and sexual behaviors, these factors may have some impact on local people’s willingness to use MCP [33]. Despite this, to some extent it can be said that in both populations there is willingness to use MCP. 

On the other hand, the non-response rate was almost double in Mozambique than in Spain (40% vs. 23.7%). There are studies that indicate that non-participation in social research could be due, among other factors, to cultural insensitivity in which some aspects of the research do not align with aspects of the culture of the participants, a lack of social support and/or societal discrimination as well as concerns about personal data [33,34,35]. This may also be associated with the message relayed by the sexual health education model (the ABC model) used within Mozambique and summarized by the slogan “abstinence, being faithful, and condom use”. Indeed, the promotion of abstinence creates social stigma around the discussion and promotion of “sexual issues” [22].

### 4.1. Policy Implications

In Spain as well as in Mozambique, participants seem concerned by side effects. This is important information for researchers, pharmaceutical companies, regulation authorities, and policymakers to pay particular attention to the side effects of any male contraceptive pill they may put in the market. The findings from previous and the current study suggest that provisions of a male contraceptive pill in low middle income countries such as Mozambique will need to be subsidized by the government, since the cost would likely be a matter of concern [14]. In addition, the use of male contraceptive pill promotion must be accompanied by counselling programs stressing the efficacy of the contraceptive pill and how to mitigate possible negative side effects [32,36]. Finally, the development of educational programs to promote more equitable roles in regard to sex and gender (e.g., responsibility of both partners for contraception) will provide populations, especially in Mozambique, with the social and educational support to improve sexual health decision making, and potentially increase men’s willingness to use MCP [31,32,34,37]. These programs would also have to include potential social and health risks due to the perception that condoms will be used exclusively to prevent STIs, as seen in Mozambique [19]. With the introduction and promotion of MCP educational programs should continue to promote condom usage in non-exclusive sexual encounters to address a possible increase in STIs; addressing stigma and de-gendering the responsibility of all contraceptive usage. Demand for these methods is therefore likely to increase when public awareness of sexual and reproductive health equity is achieved through education and culture in diverse populations [32]. 

It is also valuable to note that social perception holds more power than sexual health and safety within Mozambique, meaning inclusive sexual health education is imperative to the success of future sexual and reproductive health programs. There is evidence that approaches trying to engage men into sexual and reproductive health are most effective when they work at the personal, social, structural, and cultural levels [36]. While addressing specific life stages, and reflecting a broad and inclusive approach to sexuality, masculinities, and gender.

Research suggests there is more security and choice among Spanish population in regards to contraception compared to the Mozambican population, where women do not have limited choice when it comes to contraception solutions [17]. However, despite the ease of access to contraception in Spain [11], there are high rates of STIs that are increasing due to the decrease in the use of condom usage [38]. In this regard, Spain needs to learn from Mozambique in order to increase the number of barrier contraceptive methods used, STI testing and ensure safe sex among its population. Mozambique should begin to support policies in which traditional male roles are disseminated and reconstructed in order to promote gender equity within communities. Despite the differences found in the two countries, both countries need to empower populations during reproductive age with adequate policies aiming to achieve equity in sexual and reproductive health.

### 4.2. Limitations

The data used in the presented study did not come from samples for which a strict randomly techniques of participants selection was applied. Thus, to what extend the population recruited in Spain and in Mozambique are representative of both countries is unknown. This means that the generalization of this study results should be made with caution.

### 4.3. Strength

Compared to the previous studies the strength of the current study lies in its comparative design [14,18]. The present study also brings information on the differences regarding the willingness to use MCP between Mozambican and Spanish male participants. Shedding a light on the cultural foundation of these underlying these differences.

## 5. Conclusions

Disparities related to unintended pregnancy and abortion among low-, middle-, and high-income countries indicate the need for further action to achieve global equity in sexual and reproductive health. Continued investment is needed to ensure access to the full range of high-quality sexual and reproductive health services. Include comprehensive, inclusive package of sexual and reproductive health services, including contraception and safe abortion services in national health systems.

For shared responsibility to become a supported lifestyle choice, it necessary for men and women to have access to a wide range of contraceptives, resources, and an inclusive, comprehensive sexual education foundation that promotes equity, safety and choice. Supported by service providers, governing bodies, and communities to bring about a change in norms around contraception and gender. With this, equitable reproductive rights can be promoted among the global population.

Together with technological change, an ideological-social change in gender roles is required to guarantee equity in contraceptive and social responsibility.

## Figures and Tables

**Figure 1 ijerph-20-03404-f001:**
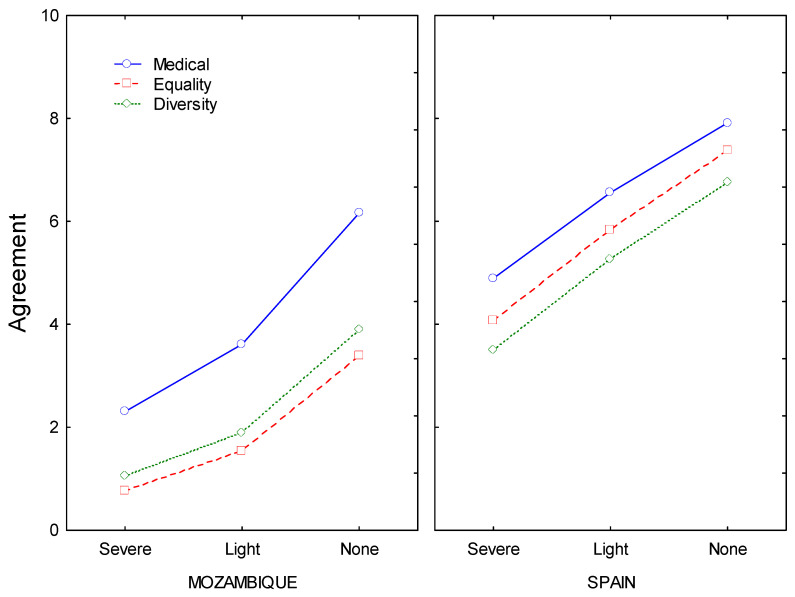
The main patterns of data that correspond to the comparison Spain-Mozambique in each panel, willingness mean scores to use the pills is on the vertical axis, and the three levels of the side effects factor are on the horizontal axis. The three curves correspond to the three levels of the context factor. In this figure, the word “Diversity” means “diversification” of the contraception method.

**Table 1 ijerph-20-03404-t001:** Sociodemographic characteristics Spain vs. Mozambique.

	Spain	Mozambique
Population	47 million of inhabitants	31 million of inhabitants
Annual average economic income per capita (GDP per capita)	25,410 €	453 €
Life expectancy at birth, total (years)	82	61
Risk of poverty	21%	46.1%
Male literacy rate	98.9%	72.6%
Unwanted pregnancy rate	11%	57%
Fertility rate	1.19 children per woman	4.78 children per woman
Statistical performance indicators (SPI): Overall score (scale 0–100)	88.9	56.2
Gender gap *	79.5% (8th country in equality)	72.3% (56th country in equality)
Human Capital Index (HCI) (scale 0–1) **	0.7 (25th country ranking)	0.4 (181st country ranking)

* Gender gap: Measures the size of the gap of said gender inequality in participation in the economy and the qualified world of work, in politics, access to education and life expectancy. ** Human Capital Index: Takes into account three variables: long and healthy life, knowledge, and decent standard of living (United Nations).

**Table 2 ijerph-20-03404-t002:** Sociodemographic characteristics of the samples of Spain and Mozambique.

Characteristic	Spain N (%)	Mozambique N (%)
Age, years		
15–20	43 (10.7)	49 (11.9)
21–25	72 (17.9)	97 (23.5)
26–30	79 (19.7)	83 (20.1)
31–35	91 (22.6)	65 (15.9)
36–40	65 (16.2)	36 (8.7)
+40	48 (11.9)	66 (16)
Unknown	4 (1)	16 (3.9)
Socioeconomic level		
<€1000/Low	99 (24.6)	209 (50.7)
€1000–1500/Intermediate	109 (27.1)	146 (35.4)
>€1500/High	116 (28.9)	41 (10)
Unknown	78 (19.4)	16 (3.9)
Education		
Primary or Secondary/<12 years	117 (29.1)	83 (20.1)
Professional/12 years	129 (32.1)	159 (38.6)
University/>12 years	156 (38.8)	154 (37.4)
Unknown	0 (0)	16 (3.9)
Religious Tradition/Religion		
Catholic/Christian	177 (44)	231 (56.1)
Muslim	10 (2.6)	76 (18.4)
Animist	0 (0)	77 (18.7)
Atheist	167 (41.6)	12 (2.9)
Jehovah	13 (3.2)	0 (0)
Protestant	7 (1.7)	0 (0)
Orthodox	7 (1.7)	0 (0)
Unknown	21 (5.2)	16 (3.9)
Religiosity		
None	183 (45.5)	0 (0)
Low	100 (24.9)	139 (33.7)
Intermediate	49 (12.1)	128 (31.1)
Strong	70 (17.5)	145 (35.2)
Total	402	412

Socioeconomic level in Spain was trichotomized: less than €1000, between €1000 and €1500 or more than €1500; in Mozambique was trichotomized in level: low, intermediate or high. Education in Spain was trichotomized: primary or secondary, professional or university; in Mozambique was trichotomized: less than 12 years, 12 years or more than 12 years.

**Table 3 ijerph-20-03404-t003:** Spain-Mozambique Comparison: Means and Standard Deviations Observed for Each Level of Each Factor.

	Factors			Spain		Mozambique	
Cost	Efficacy	Context	Side-Effects	Mean	SD	Mean	SD
30 €	99%	medical	none	9.51	2.48	7.20	2.68
30 €	99%	medical	light	8.10	2.90	4.51	2.42
30 €	99%	medical	severe	6.59	3.45	3.10	2.19
30 €	99%	equity	none	9.04	2.65	4.24	2.63
30 €	99%	equity	light	7.26	2.86	2.32	1.49
30 €	99%	equity	severe	5.74	3.38	1.63	1.10
30 €	99%	condom	none	8.22	3.08	4.88	2.76
30 €	99%	condom	light	6.54	3.05	2.84	1.78
30 €	99%	condom	severe	4.99	3.24	1.93	1.43
30 €	95%	medical	none	8.13	2.96	5.93	2.50
30 €	95%	medical	light	6.84	3.03	3.80	2.22
30 €	95%	medical	severe	5.39	3.37	2.66	1.93
30 €	95%	equity	none	7.47	2.95	3.64	2.38
30 €	95%	equity	light	6.20	2.98	2.01	1.37
30 €	95%	equity	severe	4.46	3.21	1.54	1.04
30 €	95%	condom	none	6.97	3.07	3.96	2.48
30 €	95%	condom	light	5.61	3.03	2.46	3.51
30 €	95%	condom	severe	4.05	3.06	1.73	1.31
Free	99%	medical	none	9.77	2.44	8.58	2.46
Free	99%	medical	light	8.28	2.95	5.65	2.50
Free	99%	medical	severe	6.29	3.51	4.09	2.30
Free	99%	equity	none	9.37	2.56	5.23	2.78
Free	99%	equity	light	7.62	2.91	3.13	1.78
Free	99%	equity	severe	5.51	3.41	2.06	1.37
Free	99%	condom	none	8.80	2.95	5.92	2.90
Free	99%	condom	light	7.13	3.18	3.57	2.03
Free	99%	condom	severe	4.86	3.28	2.43	1.70
Free	95%	medical	none	8.26	3.02	6.93	2.37
Free	95%	medical	light	7.04	3.24	4.49	2.29
Free	95%	medical	severe	5.33	3.34	3.37	2.21
Free	95%	equity	none	7.67	2.97	4.49	2.53
Free	95%	equity	light	6.39	3.06	2.68	1.68
Free	95%	equity	severe	4.65	3.12	1.88	1.23
Free	95%	condom	none	7.14	3.19	4.82	2.54
Free	95%	condom	light	5.88	3.16	2.88	1.78
Free	95%	condom	severe	4.10	3.06	2.10	1.54

SD: standard deviation.

**Table 4 ijerph-20-03404-t004:** Results of One-way ANOVA: The effects of the Country on the Factor-levels.

Factor-Level	df	MS	*F*	*p*	η^2^_p_
Cost					
30 €	1	102.486	42.6	<0.001	0.56
Free	1	68.89	22.72	<0.001	0.40
Efficacy					
99%	1	101.09	33.0	<0.001	0.49
95%	1	70.04	34.6	<0.001	0.50
Context					
Medical	1	35.6	12.97	0.0016	0.37
Feminism	1	90.22	45.17	<0.0001	0.67
Condom	1	50.39	24.17	<0.0001	0.52
Side-effects					
None	1	49.731	32.24	<0.0001	0.59
Light	1	75.468	81.1	<0.0001	0.79
Severe	1	46.575	73.47	<0.0001	0.77

Df: degree of freedom; MS: mean square, *F*: variance ratio value; *p*: probability value (at 0.05 significance level); η^2^_p_: effect size.

## Data Availability

The data supporting this finding are available at: https://osf.io/mct2z?show=view&view_only= (accessed on 22 August 2022). The material used in this study is available from the corresponding author upon reasonable request.

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
