# Peer review of "Willingness to Use Male Contraceptive Pill: Spain-Mozambique Comparison"

_ijerph, 2023, doi:10.3390/ijerph20043404_

Round 1

Reviewer 1 Report

This manuscript  compared the level of willingness to take a male contraceptive pill between two population samples (Spain = 402 participants; Mozambique = 412 participants). One-way ANOVAs were performed comparing the average scores of Mozambique and Spain at The cost of the pills, Efficacy, Side effects (none, mild and severe), and Context (disease, condom abandonment, and 18 diversification of contraceptive methods).

I have the comments below:

1. The title could be more precise and less misleading. Because the sample from Spain was collected in the autonomous community of Aragon instead of the whole country, the title may be changed to "Theoretical Acceptability of a Male Contraceptive Pill: Comparison between Mozambique and the Aragon Autonomous Region in Spain"

2. The conclusion needs to be expanded. Are there any policy implications?

This manuscript require further proofreading before the next submission. There are gramatical error across all sections.

Reviewer 2 Report

The article concerns the actual problem of attaining gender equality in birth control methods. Males’ attitudes and believes; gender ideology play crucial role in acceptance and willingness of taking contraceptive pills. The authors focus on cross-cultural dimension of the problem comparing the data gained from Spain and Mozambique group of the respondents, and this choice is stemmed from the idea of global North/South opposition. Mozambique gives interesting comparison with Spain, because this region shows the highest increase in spread of contraceptive practice, and it is interesting if this practice may be steadily widened for men and due to what factors.

Both groups of respondents are quite comparable in number and age. The same measurement tool was used in both countries. Four factors were chosen as variables of decision making to take a male contraception pill: its cost, efficacy, side effects, and the context of use (I’d rather call it reasons to take pills, but let it be context). The statistical procedure is applicable to the data gained and allows one to achieve the stated objective. The very design of research appears interesting and is sound. But the report is poor.

Within Introduction section, there is no theoretical model and description of the basic notion of acceptability. In the text, the authors replace this term with another: willingness. I am not sure that the acceptability of taking pills and the willingness to pursue birth control by pills are the same phenomena. If so, it takes to be revealed in the theoretical grounds. The model may approve the choice of variables for empirical research. And if you do not describe any model, why is the article entitled Theoretical Acceptability of…? There is no any theory of acceptability in the text. What if the data gained relate to decision making model, not willingness to take pills nor acceptability to regulate conception in this way? It is not clear from Introduction. In addition, if we speak about men’s decision, willingness, or acceptability to be involved into birth control practices as direct agents/doers who moderate their own hormonal functioning, there are also models considering the impact of gender ideology, masculinity ideology (J. Pleck), etc. I do not mean that the authors should apply to gender studies, but to describe some theoretical mechanisms and refine the terms used is necessary.

Before getting acquainted with the factors for taking or not male contraceptive pill, it would be helpful to catch the sense of a “new revolution” in the contraception field due to the Introductory section of the article – shortly, what are the mechanisms, impacts, effects, etc. What information about the medicines was provided to the respondents before questioning?

It is not enough to refer to another article when you tell about the recruitment of the Spanish group. It should be clearly expressed comparability or incompatibility of the sampling procedures in the text.

No research can be aimed at comparing, since comparing is a procedure, not a result of a study. This research might be aimed, for instance, at revealing the specific structure of willingness, or acceptability, or decision-making process in Spanish and Mozambican males. Or it might be aimed at describing the hierarchies of the major factors that have a strong impact on men’s willingness to take contraceptive pills within the global opposition of the North / South or under different sociocultural conditions in Spain and Mozambique. The aim should follow from the content of the theoretical analysis and model.

In Discussion section, empirical data are considered through the concepts of gender role beliefs, social representations of contraception, differences in sexual health education in Spain and Mozambique, and other things. But there is nothing of these in theoretical explanation of the study done in the Introduction. The authors suggest the specific elements for sexual health education programmes to improve the acceptability of male contraceptive pills and reduce other risks (of STDs infections, etc.), which do not meet the content of the research done (there is nothing about perception of condoms, propaganda of progressive gender roles in main sections of the text).

The text also contains some inadequacies:

1.       Table 1 has incorrect usage of terms. There should be used the term Fertility rate (is calculated by the number of children born to women in a population, and this number is presented in the Table), whereas Fecundity rate is measured by assessing the number of gametes (sperm and egg) a person can possess. I am aware that the authors do not apply to the statistics of gametes within the individuals of both countries.

2.       I would recommend changing the title of Table 2 to make clear its content:  willingness to use male contraceptive pill under different conditions. But it is not clear why willingness, not acceptability?

3.       It is unclear what the context of (pill) use really consists of: according to line 105 this factor concerns medical reason, reasons for equity in the couple, to stop using condom as contraception; in line 133 we read that this factor concerns medical condition, female partner, and diversity of methods. It is necessary to follow one comprehensive description of the context. Further, it is unclear why the authors use only diversity (of what?) to demonstrate the statistics for the context factor in Figure 1. Where are medical reasons, reasons for equity in the couple, to stop using condom as contraception?

4.       There is no description of the data presented in Table 2: what is the meaning of dispersions, which interpretations of those values obtained could be suggested, how the willingness (or acceptability?) parameters are manifested within the Spanish and Mozambican samples? The table should be described.

Else, I would clarify keywords: for instance, ‘acceptability of male contraceptive pills’,willingness to take contraceptive pills’, etc. according with the sense of research. The keywords given are empty and poor.

There are many misprints in the text given:

Line 2:  Theorical – theoretical.

Line 13: compering – comparing.

Line 36: male contraceptive male – male contraceptive pill.

I found the passage at lines 12-13 unfinished:  which (verb? is?) in relatively advanced development stage.

The article may be published after a revision.

Reviewer 3 Report

Dear Authors,

you have presented interesting topic in your manuscript, but I have some minor and major comments about it.

Abstract
What does it mean MCP in Abstract - all abbreviations should be defined the first time they appear in each of sections, as well as in the abstract.

The references in the main text should be in [].

Methods
This section is not clear presented - it has to be modified.
In line 76 and 77 there are references, could you explain what for?
Explain how did you do the study? What were the inclusion and exclusion criteria for study?
In Statistical analysis you present that the significance threshold was set at 0.001, but under the Table 3 at .05 significance level - it is not clear?

Results
At the beginnig of section result should be presented characteristics of the study group.
Description of table 3 should be above the Table.

Add information about strength of your study. What new adds this study while you done two studies indepently about Spain and Mozambique?

Section References have to be modified as to Instruction for Authors.
